# Land Cover Classification with Multispectral LiDAR Based on Multi-Scale Spatial and Spectral Feature Selection

**Shuo Shi [1], Sifu Bi [1,*], Wei Gong [1,2], Biwu Chen [3], Bowen Chen [1,4], Xingtao Tang [1], Fangfang Qu [1] and Shalei Song [5]**

1. State Key Laboratory of Information Engineering in Surveying, Mapping and Remote Sensing, Wuhan University, 129 Luoyu Road, Wuhan 430072, China; shishuo@whu.edu.cn (S.S.); weigong@whu.edu.cn (W.G.); chenbowen1204@whu.edu.cn (B.C.); whutxt@whu.edu.cn (X.T.); fangqu@whu.edu.cn (F.Q.)
2. Electronic Information School, Wuhan University, Wuhan 430072, China
3. Shanghai Radio Equipment Research Institute, Shanghai 201109, China; cbw_think@whu.edu.cn
4. Chinese Antarctic Center of Surveying and Mapping, Wuhan University, Wuhan 430079, China
5. State Key Laboratory of Magnetic Resonance and Atomic and Molecular Physics, Innovation Academy for Precision Measurement Science and Technology, Chinese Academy of Sciences, Wuhan 430071, China; songshalei@apm.ac.cn
* Correspondence: bisifu@whu.edu.cn; Tel.: +86-132-5151-8103

**Abstract:** The distribution of land cover has an important impact on climate, environment, and public policy planning. The Optech Titan multispectral LiDAR system provides new opportunities and challenges for land cover classification, but the better application of spectral and spatial information of multispectral LiDAR data is a problem to be solved. Therefore, we propose a land cover classification method based on multi-scale spatial and spectral feature selection. The public data set of Tobermory Port collected by the Optech Titan multispectral airborne laser scanner was used as research data, and the data was manually divided into eight categories. The method flow is divided into four steps: neighborhood point selection, spatial–spectral feature extraction, feature selection, and classification. First, the K-nearest neighborhood is used to select the neighborhood points for the multispectral LiDAR point cloud data. Additionally, the spatial and spectral features under the multi-scale neighborhood (K = 20, 50, 100, 150) are extracted. The Equalizer Optimization algorithm is used to perform feature selection on multi-scale neighborhood spatial–spectral features, and a feature subset is obtained. Finally, the feature subset is input into the support vector machine (SVM) classifier for training. Using only small training samples (about 0.5% of the total data) to train the SVM classifier, 91.99% overall accuracy (OA), 93.41% average accuracy (AA) and 0.89 kappa coefficient were obtained in study area. Compared with the original information's classification result, the OA, AA and kappa coefficient increased by 15.66%, 8.7% and 0.19, respectively. The results show that the constructed spatial–spectral features and the application of the Equalizer Optimization algorithm for feature selection are effective in land cover classification with Titan multispectral LiDAR point data.

**Keywords:** land cover classification; multispectral LiDAR; Titan laser scanner; multi-scale neighborhood features; equalization optimizer; feature selection

## 1. Introduction

Land cover classification is an important reference basis for public policy planning, Earth resources' management and climate monitoring [1]. Because of the large-scale characteristics of remote sensing technology, it has been widely used in land cover classification. In the past, passive remote sensing images have obtained excellent land cover classification results by virtue of their rich spectral information. The LiDAR sensor can obtain the 3-D space information of the objects and can classify the objects by the height and position information. However, with the increasing demand of land cover classification, the image

classification technology of multi-sensor data fusion has become a hot research topic [2,3]. The spectral information of passive remote sensing images is complementary to the spatial information of LiDAR point clouds, and the data fusion strategy has achieved better results in land cover classification [4,5]. Moreover, the addition of highly informative information is a huge boost to classification [6–8]. However, the prerequisite for the fusion of LiDAR data and remote sensing data for land cover classification is that precise registration and uniform resolution must be carried out. This work is difficult to achieve accurately.

As a new type of sensor, multispectral LiDAR can obtain the information of multi-wavelength, as well as 3-D space, which solves the problem of precise registration of passive remote sensing data and LiDAR data from the hardware level and provides powerful data support for classification. Past researchers have designed different multispectral LiDAR systems. Gong et al. [9,10] successfully developed a four-wavelength (556 nm, 670 nm, 700 nm and 780 nm) ground observation multispectral LiDAR system for remote sensing classification and monitoring of vegetation. Similarly, Woodhouse et al. [11], Wallace et al. [12] and Niu et al. [13] respectively developed a set of four-wavelength multispectral LiDAR systems for vegetation information extraction. As an active remote sensing technology, multispectral LiDAR can obtain the spatial and spectral information of the objects without external interference, and finally get the multispectral point cloud. The above studies are for laboratory scenarios, and the land cover classification work needs to be supported by airborne data. Titan of Teledyne Optech is the first commercial airborne multispectral LiDAR sensor with three separate active imaging channels (1550 nm, 1064 nm, 532 nm) capable of obtaining spatial and spectral information simultaneously. Since each channel works independently, the Titan multispectral LiDAR sensor is not strictly a multispectral LiDAR. However, in the past research, Titan multispectral LiDAR data still shows great potential for land cover classification [14–17]. Compared to Titan's single-channel point clouds, multi-channel point clouds show a significant improvement in land cover classification [18–20].

At present, the application method for Titan multispectral LiDAR data is still in the exploratory stage. In the land cover classification work, its classification methods can be divided into image-based and point-based. The image-based land cover classification work is to rasterize the spectral information and height information of the point cloud data of the Titan multispectral LiDAR separately and classify the land use types on the image. This classification method converts 3-D data into 2-D data, reduces the complexity of data processing, and is suitable for large-area land cover classification work. A lot of work in the past was carried out in this scenario and can obtain classification accuracy of more than 90% [18,21–24]. However, the rasterized image will cause the loss of spatial information and spectral information. Some objects (such as the ground under vegetation cover, fences, power lines, etc.) are degraded, making it difficult to classify them. In addition, existing studies have pointed out that the point-based classification effect is better than the image-based classification effect [16,19]. Therefore, this research will be based on the Titan multispectral LiDAR point cloud to carry out land cover classification in a 3-D point cloud scene.

To make full use of the integrated spatial–spectral information of multispectral LiDAR, reasonable and effective feature extraction is necessary [18,22–24]. At present, there is little research on point-based land cover classification for Titan multispectral LiDAR point cloud data [14,19,25]. On the Titan multispectral LiDAR point cloud data, Wichmann et al. pioneered land cover classification based on point clouds [14]. They focused on the spectral feature of specific objects to explore the potential of multispectral LiDAR point clouds in land cover classification. Ekhtari et al. [19] use point cloud data to carry out land cover classification work, divide the point cloud into single-return points and multi-return points for separate processing. Single-return points are classified by spectral intensity and height information, and multi-return points are used for height information and neighborhood statistical information classification, and finally achieved better results. Wang et al. [25] proposed a three-dimensional land cover mapping point cloud classification model based

on tensor representation, which uses multispectral points and their neighborhood points to represent each point as a second-order tensor, and introduces the TMDE algorithm to obtain low-dimensional spatial and spectral features which are used for subsequent classification and indicate that the spatial neighborhood information has a key role in classification. Existing research has shifted from the primary spectral intensity features to the development of deeper features, especially the introduction of neighborhood information for feature extraction. However, the currently applied neighborhood information is only on a single scale, and the classification performance of multi-scale neighborhood information has not been explored. In this study, the method of neighborhood point selection is used to extract the spatial–spectral neighborhood features of multiple scales, and the features are cascaded to obtain the multi-scale neighborhood spatial–spectral features of the multispectral LiDAR. However, the cascade of multi-scale neighborhood features will make the dimensionality of the features too high, causing the "Hughes" phenomenon, resulting in the classification accuracy not increasing, but decreasing [26].

Aiming at the problem of Titan multispectral LiDAR data in land cover classification, a land cover classification method based on multi-scale neighborhood feature selection is proposed. The multi-scale neighborhood is used to extract the spatial spectrum feature information of the multispectral LiDAR point cloud data and give full play to the spatial spectrum synergy. The Equalization Optimizer algorithm is used for feature selection to reduce the feature dimensions and avoid the occurrence of the "Hughes" phenomenon. Finally, feature subset with the best classification effect is obtained, and input into the supervised classifier for classification. The contribution of the research is mainly divided into three aspects:

- The use of multi-neighborhood scales for the selection of neighborhood points.
- Among the existing classification features, the spectral feature and spatial feature of Titan multispectral LiDAR are extended.
- Aiming at the high dimensionality caused by multi-scale neighborhood feature cascade, the Equalization Optimizer algorithm is used for feature selection.

## 2. Materials

### 2.1. Multispectral LiDAR Data Acquisition

The multispectral point cloud data used in this study was collected by the Optech Titan multispectral LiDAR system. Titan multispectral LiDAR is the first commercial airborne multispectral sensor, including 3 wavelength LiDAR channels, respectively 1550, 1064, and 532 nm (Table 1). Each spectral channel records the echo separately, and each point cloud only contains the spectral intensity value of one channel.

**Table 1.** Optech Titan multispectral LiDAR data acquisition system sensor channel specifications and experimental data characteristics.

| Items | Channel 1 | Channel 2 | Channel 3 |
|---|---|---|---|
| Laser Wavelength | 1550 nm | 1064 nm | 532 nm |
| Beam divergence | $\approx$0.36 m Rad | $\approx$0.3 m Rad | $\approx$1 m Rad |
| Look angle | 3.5 degrees forward | 0 degrees | 7.0 degrees forward |
| Pulse repletion frequency | 50–300 kHz | 50300 kHz | 50–300 kHz |
| Intensity quantization level | 12bit | 12bit | 12bit |
| Point density | 19.8 pts/m$^2$ | 20.8 pts/m$^2$ | 21.2 pts/m$^2$ |
| Ground pulse footprint size | $\approx$0.18 m | $\approx$0.15 m | $\approx$0.5 m |
| Mean flying height | 500 m | 500 m | 500 m |

Multispectral LiDAR point cloud data were collected from Tobermory Port, Canada, and the size of 211 $\times$ 109 m$^2$ is intercepted as the research area, with a total of 1,691,297 point cloud data. The area includes trees, buildings, roads, impervious ground, grassland, unused land, cars, and power lines. Traditional single-wavelength LiDAR is difficult to distinguish objects with similar elevation information such as roads, grasslands, and

impervious ground, and 2-D images cannot distinguish power lines, under-forest objects, etc. Therefore, according to the characteristics of 3-D point cloud data and the standard of land use type, referring to the RGB image of Google Maps platform, the multispectral point cloud is divided into Tree, Building, Road, Impervious ground, Grassland, Unused land, Car and Power line (Figure 1). It should be noted that the land use type focuses on social attributes, and land cover type focuses on natural attributes. In past research, the boundaries between the two have become blurred [15,19,21,25]. For the convenience of presentation, only "land cover" is used in this article.

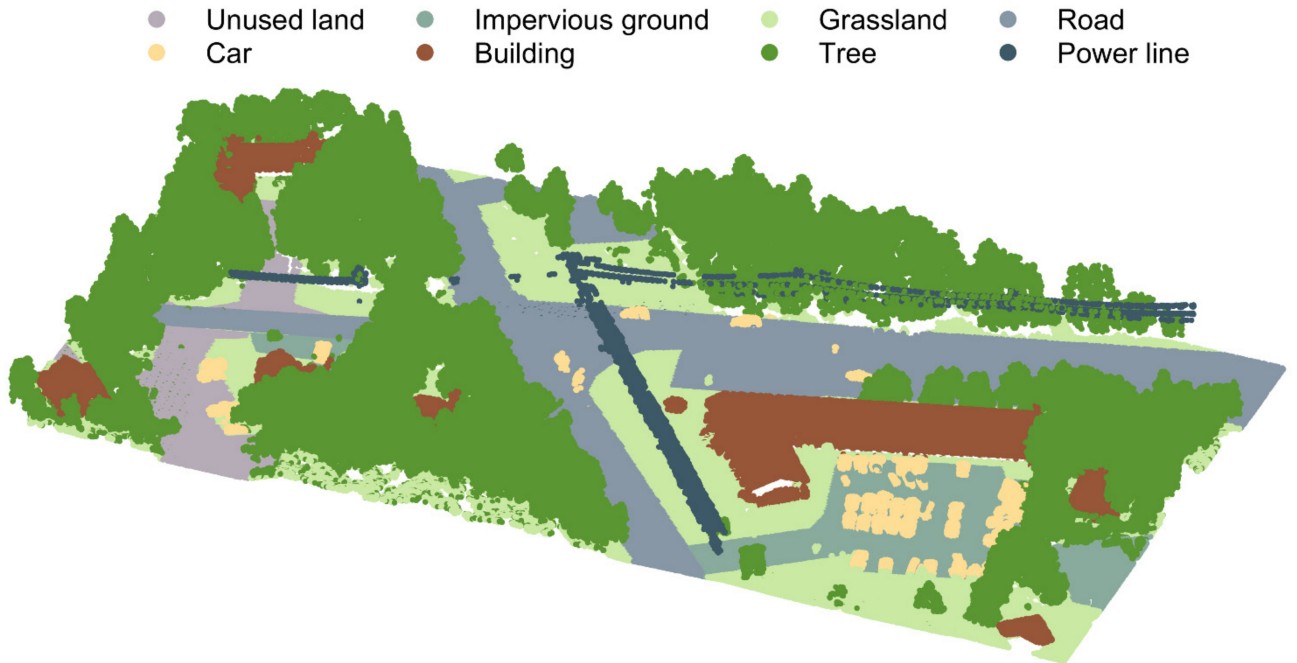

**Figure 1.** Multispectral LiDAR point cloud rendered by ground truth value.

### 2.2. Data Preprocessing

Based on the characteristics of Titan multispectral LiDAR data, that is, 3 channels generate independent point cloud files, each channel point cloud data only contains the spectral intensity value of the channel. The land cover classification based on raster images can use the spectral intensity of each channel for interpolation processing, and the land cover classification based on point cloud requires special data preprocessing according to the characteristics of the data. To obtain true multispectral LiDAR point cloud data, Wichmann et al. [27] developed a tool to search for point clouds of other channels within a 1 m radius for each point cloud, and assign its spectral intensity value to the central point cloud. The spectral intensity of a channel that does not exist in a radius of 1 m is assigned to 0, so that all point clouds obtain the spectral intensity values of 3 channels, including a true value and 2 values obtained by processing. Morsy et al. [23] also adopted a similar method. Ekhtari et al. [19] used a neighborhood radius of 2 m and eliminated the point cloud that does not have neighborhood points on a channel. However, Wang et al. [25] only used the point cloud of the 1550 nm channel as a reference and supplemented the spectral intensity information by searching for the nearest neighbors of other channels.

In order to maximize the use of the original spatial information and spectral information, we introduced the interpolation method, which improved the method of Wichmann et al. [14]. Based on the assumption that there is a correlation between the spectral intensities of neighboring points, this study retains the point clouds of all channels and uses the intensity information of the neighboring points of other channels to assign the intensity to each single-wavelength point cloud. Take the point cloud of channel 1 as an example, search for the 5 channel 2 point clouds of each point cloud nearest neighbors, and

remove the neighboring points with a distance greater than 1 m. Then, use the inverse distance weighted interpolation method to assign the point cloud of channel 1 according to the spectral intensity of the point cloud of channel 2, and eliminate the point cloud with no neighbor points within 1 m. The intensity assignment of channel 2 and channel 3 is the same. Finally, point cloud data with multi-wavelength spectral intensity is obtained.

Since the calculation of spectral features is based on spectral reflectance, it is necessary to convert the intensity value of each channel into pseudo reflectance. Refer to the processing method of Wichmann et al. [14]; take the 99th quantile of the spectral intensity of each channel as the reflectance spectral intensity of the reference plate, and adjust the reflectance to 1 if the reflectance exceeds 1. Finally, point cloud data with multispectral pseudo-reflectivity is obtained (Figure 2). Among them, road indicator lines and objects such as cars have spectral abnormalities due to specular reflection.

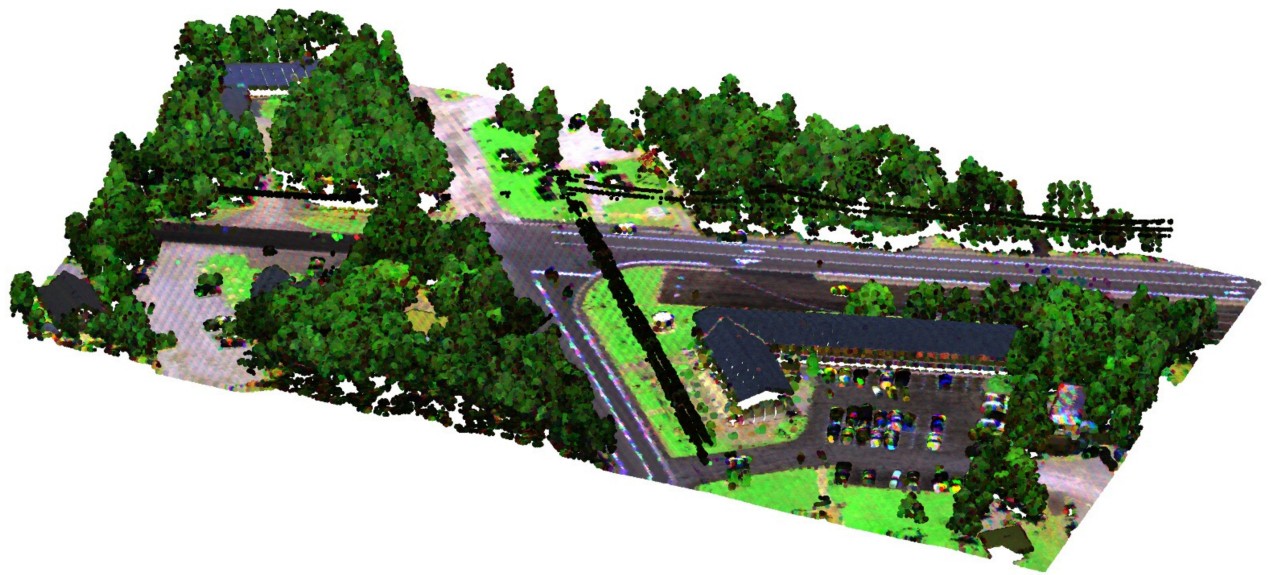

**Figure 2.** Multispectral LiDAR point cloud rendered by pseudo-color (R: channel 1, G: channel 2, B: channel 3).

### 2.3. Training and Testing Samples

It is time-consuming and laborious to manually classify the LiDAR point cloud. In the actual land cover classification work, it is unrealistic to use a large number of training samples for classifier training. Therefore, we expect to be able to obtain good classification accuracy by using fewer training samples. After the multispectral LiDAR point cloud is preprocessed, the area also contains 1,627,877 point clouds, which are divided into 8 categories. We randomly selected 1000 samples for each type of object according to the category, and a total of 8000 samples were used as the training samples, and the remaining 1,619,877 point clouds were used as the testing samples (Table 2).

**Table 2.** Number of training and testing samples.

| Land Cover Types | Testing Samples Size (#) | Training Samples Size (#) |
| --- | --- | --- |
| Tree | 639,951 | 1000 |
| Building | 123,757 | 1000 |
| Road | 276,651 | 1000 |
| Impervious ground | 117,829 | 1000 |
| Grassland | 385,786 | 1000 |
| Unuse land | 52,019 | 1000 |
| Car | 16,559 | 1000 |
| Power line | 7325 | 1000 |

## 3. Methods

The proposed method for land cover classification using Titan multispectral LiDAR data is presented in Figure 3. This method is based on pre-processed Titan Multispectral LiDAR data. Firstly, about multiple scales, the selection of neighborhood points is performed, and the spatial and spectral features of the multi-scale neighborhood are extracted. Serializing them to obtain high-dimensional spatial–spectral features, but directly using high-dimensional features has poor performance and high computational cost. Therefore, the feature selection algorithm of the Equalization Optimizer is used to obtain a low-dimensional multi-scale neighborhood spatial–spectral feature subset. Finally, feature subset is input into the SVM classifier for classification, and the classification results are evaluated according to the ground truth value. The details of method are presented in the following subsections.

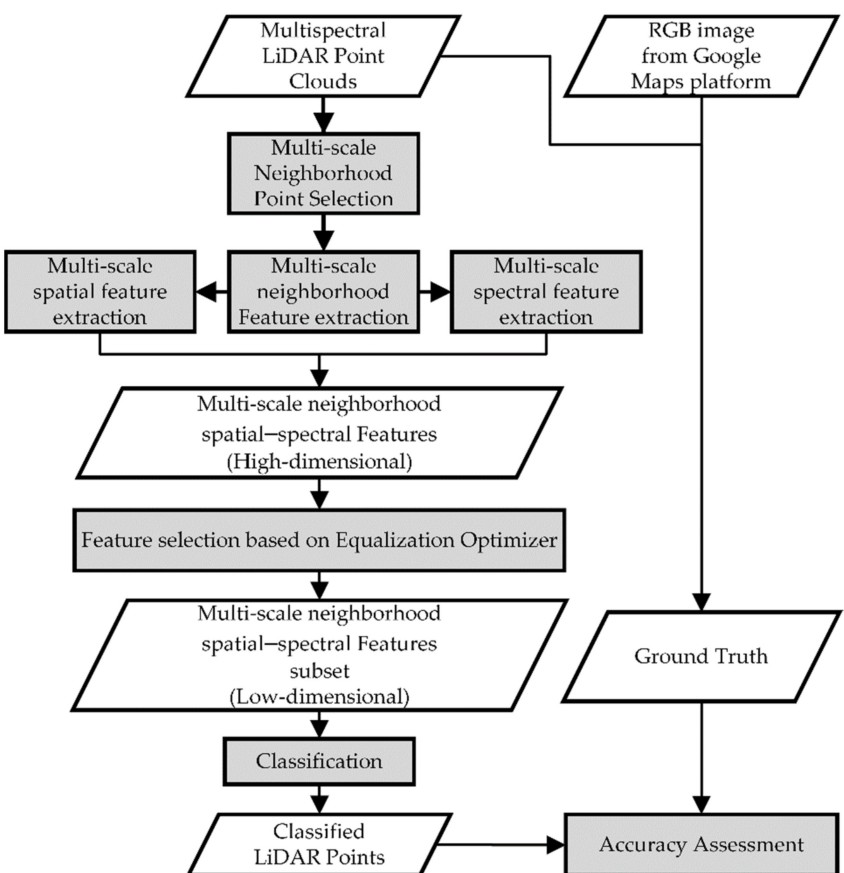

**Figure 3.** The overall classification method.

### 3.1. Neighborhood Point Selection

The classification framework of the research application is improved from the single-wavelength LiDAR point cloud classification framework [28]. In the classification framework, the selection of neighborhood points will affect the utilization of original information and the effect of feature extraction, so the selection method of neighborhood points is also the focus of research [27–30]. Commonly used neighborhood types include spherical neighborhood, cylindrical neighborhood, and K-nearest neighborhood. For a given point cloud P, the spherical neighborhood includes all point clouds whose distance from the P is less than the radius R. The cylindrical neighborhood includes all point clouds with a radius less than R from the P on the ground projection point. The K-nearest neighborhood includes the K point clouds closest to the P [30]. According to traditional neighborhood types, such as cylindrical neighborhoods, Wang et al. [29] improved them and achieved better performance in the classification of power lines in urban areas. These three types

of neighborhoods all rely on empirical or heuristically given radius or K values, which are often different in different scenarios. To solve this problem, the idea of the adaptive neighborhood is proposed, including surface variation, dimensionality-based, and eigenentropy-based neighborhood point selection [27,31,32].

Due to different scenarios, the optimal neighborhood scale is also different. Although the adaptive neighborhood can select the optimal neighborhood-scale based on local features, the computational cost and time cost of point-by-point search are very expensive. In addition, the neighborhood selection method proposed above is only feature extraction at a single-scale. As far as we know, there is no research on land cover classification using multi-scale neighborhood features on Titan multispectral LiDAR data [25]. Therefore, the study uses multi-scale to extract spatial features and spectral features of multispectral LiDAR point cloud. Due to the uneven spatial density of the point cloud of Titan data, the use of spherical neighborhoods or cylindrical neighborhoods cannot guarantee the number of neighborhood points, and there are situations where spatial features cannot be extracted, so K-nearest neighborhoods are used for feature extraction. Considering that the larger the K value, the higher the computational cost of feature extraction, it is more inclined to use a smaller K value when selecting the size of the multi-scale neighborhood. Therefore, the study used four different K values of 20, 50, 100, and 150 to extract the spatial–spectral features.

### 3.2. Multi-Scale Neighborhood Features Extraction

3.2.1. Multi-Scale Spatial Features

The multi-scale spatial feature extraction method is similar to the single-scale spatial feature extraction method. According to different K values, different spatial features are extracted in sequence, and the features are connected in series to obtain multi-scale spatial features. The spatial information of multispectral LiDAR is the same as that of single-wavelength LiDAR, including x, y, and z spatial coordinate information. Based on this, the spatial feature expression method similar to traditional LiDAR is adopted [28]. For the selected neighborhood points, a covariance matrix of local coordinates is established, and its eigenvalues and eigenvectors are calculated. Calculate the proportion of each eigenvalue to the total eigenvalues and arrange them in descending order, that is, the 3-component values of the geometric space tensor $\lambda^K_1$, $\lambda^K_2$, $\lambda^K_3$, and use them as the first 3 spatial features, where $K$ is the neighborhood scale. Before performing spatial feature calculations, the sum of the 3 feature values is standardized to $e^K_1$, $e^K_2$, $e^K_3$. Based on 3 normalized geometric tensors, construct linearity ($L^K_\lambda$), planarity ($P^K_\lambda$), scattering ($S^K_\lambda$), omnivariance ($O^K_\lambda$), anisotropy ($A^K_\lambda$), eigenentropy ($E^K_\lambda$), change of curvature ($C^K_\lambda$), there is a total of 7 spatial features. They are computed as follows:

$$L^K_\lambda = \frac{e^K_1 - e^K_2}{e_1}$$

$$P^K_\lambda = \frac{e^K_2 - e^K_3}{e_1}$$

$$S^K_\lambda = \frac{e^K_3}{e^K_1}$$

$$O^K_\lambda = \sqrt[3]{e^K_1 e^K_2 e^K_3}$$

$$A^K_\lambda = \frac{e^K_1 - e^K_3}{e^K_1}$$

$$E^K_\lambda = -\sum_{i=1}^{3} e^K_i \ln e^K_i$$

$$C^K_\lambda = \frac{e^K_3}{e^K_1 + e^K_2 + e^K_3}$$

Among them, the planarity value of flat objects is higher, and the surface objects with irregular surfaces have higher values of scattering, omnivariance and change of curvature. Verticality $V^K{}_\lambda$ is constructed using the feature vector of the third component $V = (v_1,v_2,v_3)$. It is computed as follows:

$$V^K{}_\lambda = 1 - \frac{v_3}{||V||_2} = 1 - \frac{v_3}{\left[\sum_{i=1}^{3} v_i^2\right]^{\frac{1}{2}}}$$

In addition, height-based features are extracted, such as the $Z$ coordinate value, the neighborhood height difference $\Delta z^K$, and the neighborhood height standard deviation $Std^K z$. Based on the neighborhood selection, the neighborhood radius $R^K$ and the neighborhood point cloud density feature $D^K$ are extracted. Among them, $k$ is the number of neighborhood points. They are computed as follows:

$$\Delta z^K = \max(z) - \min(z)$$

$$Std^K z = \left[\frac{1}{k}\sum_{k=1}^{k}(z - mean_z)^2\right]^{\frac{1}{2}}$$

$$R^K = max\left(\left[(x - x_i)^2 + (y - y_i)^2 + (z - z_i)^2\right]^{\frac{1}{2}}\right), \ \ i = 1, 2, \ldots, k$$

$$D^K = \frac{k}{\frac{4}{3}\pi R^3}$$

Single-scale neighborhood spatial features include 19-dimensional features. After removing the same features (such as $Z$), multi-scale neighborhood features include 73-dimensional spatial features.

### 3.2.2. Multi-Scale Spectral Features

Multi-scale spectral features are similar to multi-scale spatial features. The spectral features of a single neighborhood are first extracted and then the features are connected in series to obtain multi-scale spectral features. The spectral information of the neighborhood can reduce the influence of spectral abnormal values. Multispectral LiDAR point cloud data provides spectral information of multiple wavelengths. Titan point cloud data after preprocessing has spectral intensity information of three wavelengths, which is normalized to pseudo reflectance information using the 99th quantile of each channel. Spectral features include features constructed using the reflectivity of a single wavelength and features constructed from the interrelationship of reflectivity of different wavelengths. Among the features constructed by single-wavelength reflectance, the original reflectance information ($I_{jk}$), the average value of the neighborhood point reflectance ($mean^K{}_j$), and the normalized value of the central point cloud reflectance ($norm^K{}_j$) are used to describe the relationship between the central point cloud and the neighborhood point cloud. They are computed as follows:

$$mean^K{}_j = \sum_{k=1}^{K} \frac{I_{jk}}{K}$$

$$norm^K{}_j = \frac{I_{jk} - mean_j}{std_{dev_j}}$$

The standard deviation ($std^K{}_{dev_j}$), skewness ($skewness^K{}_j$), kurtosis ($kurtosis^K{}_j$) and coefficient of variation ($CV^K{}_j$) of the reflectance of neighboring points are used to describe the statistical characteristics of the reflectance of neighboring points. They are computed as follows:

$$std^K{}_{dev_j} = \left[\frac{1}{K}\sum_{k=1}^{K}\left(I_{jk} - mean_j\right)^2\right]^{\frac{1}{2}}$$

$$skewness^K{}_j = \frac{mean_j^3}{std_{dev_j}^3} = \frac{1}{K}\times\sum_{k=1}^{K}\frac{\left(I_{jk}-mean_j\right)^3}{std_{dev_j}^3}$$

$$kurtosis^K{}_j = \frac{mean_j^4}{std_{dev_j}^4} = \frac{1}{K}\times\sum_{k=1}^{K}\frac{\left(I_{jk}-mean_j\right)^4}{std_{dev_j}^4}$$

$$CV^K{}_j = \frac{std_{dv_j}}{mean_j}$$

The features constructed by the interrelationship of reflectance of different wavelengths include ratio and NDFI. The ratio is the ratio of the total reflectivity of each channel to the total channel reflectivity. NDFI is constructed based on the characteristics of Titan multispectral LiDAR data [22]. They are computed as follows:

$$ratio^K{}_j = \frac{mean_j}{\sum_{j=1}^{3} mean_j}$$

$$NDFI^K{}_{G-NIR} = \frac{mean_3 - mean_2}{mean_3 + mean_2}$$

$$NDFI^K{}_{G-MIR} = \frac{mean_3 - mean_1}{mean_3 + mean_1}$$

$$NDFI^K{}_{NIR-MIR} = \frac{mean_2 - mean_1}{mean_2 + mean_1}$$

Single-scale neighborhood spectral features include 24-dimensional features. After removing the same features (such as original reflectance information $I_{jk}$), multi-scale neighborhood features include 87-dimensional spectral features.

### 3.3. Feature Selection Based on Equalization Optimizer

The construction of the feature set is empirical. In order to screen out the features suitable for Titan multispectral LiDAR point cloud data in land cover classification, it is necessary to construct a high-dimensional feature set. So in Section 3.2, we obtained 160-dimensional high-dimensional spatial–spectral features. At the same time, the multi-scale neighborhood features constructed by the same definition must have redundancy, and feature selection is necessary.

Feature selection enhances the classification accuracy by selecting a subset of features, or reduces the dimension of feature sets and computational costs without reducing the classification accuracy of the classifier [33]. Feature selection can be divided into filtering feature selection and wrapping feature selection [34] according to evaluation criteria. Filtered feature selection is to evaluate features through a certain measurement index and retain a subset of features with good performance. The synergies between features are not taken into account in this process, and it is incomplete to consider only the representation of a single feature [35]. Wrapper feature selection is a problem that takes the classification results as the evaluation criteria and transforms feature selection into search optimization. Most of the wrapped feature selection algorithms are developed based on optimization algorithms, but the traditional optimization algorithms cannot deal with complex problems. Therefore, meta-heuristic algorithm is developed in feature selection. The genetic algorithm (GA) [36], particle swarm optimization (PSO) [37], simulated annealing (SA) [38] and ant colony optimization (ACO) [39] are some of the most conventional meta-heuristic approaches. They all belong to different categories of meta-heuristics, and many researchers in different fields have evaluated their performance. The equalization optimizer-based

algorithm is an optimization algorithm designed based on a physical approach and inspired by the control volume-mass balance model used to estimate dynamic and equilibrium states [40]. Through the fitting test of 58 functions, the Equalization Optimizer algorithm is significantly better than the PSO, GA, Gray Wolf Optimizer (GWO), Gravity Search Algorithm (GSA), Salp Swarm Algorithm (SSA) and Covariance Matrix Adaptive Evolution Strategy algorithm (CMA-ES).

The basic theory of the Equalization Optimizer algorithm comes from the control volume mass balance model:

$$V\frac{dC}{dt} = QC_{eq} - QC + G$$

where $V$ is the control volume, $C$ is the concentration of particles in the control volume, $V\frac{dC}{dt}$ is the rate of change in mass in the control volume, $Q$ is the volumetric flow rate into and out of the control volume, $C_{eq}$ is the concentration of particles inside the control volume at an equilibrium state without generation, and $G$ is the mass generation rate inside the control volume.

After deforming and integrating it, the basic formula for equalization optimizer can be obtained.

$$C = C_{eq} + (C_0 - C_{eq})F + \frac{G}{\lambda V}(1 - F) \tag{1}$$

$C_{eq}$ is the equilibrium pool, and the pool retains the four best individuals and one average individual selected during the iterative process. $F$ is an exponential term, which controls and balances the weight of the Equalization Optimizer algorithm in the exploration and exploitation process. $G$ represents the generation rate, which is used for the update rate of each individual concentration value. For the detailed calculation methods of $C_{eq}$, $F$ and $G$, refer to the research of Faramarzi et al. [40]. After the items are introduced, the Formula (1) is expressed as Formula (2), where $V$ is set to the unit value 1.

$$\vec{C} = \vec{C_{eq}} + \left(\vec{C} - \vec{C_{eq}}\right)\vec{F} + \frac{\vec{G}}{\lambda V}\left(1 - \vec{F}\right) \tag{2}$$

Since the original Equalization Optimizer algorithm is designed to solve the continuous optimization problem, the feature selection algorithm is often based on the binary method, that is, the feature is selected and removed, and the original Equalization Optimizer algorithm needs to be improved. The improvements made are in two aspects, including the calculation of fitness values and the selection strategy of features.

Based on the purpose of feature selection, that is, to reduce feature dimensions and improve classification accuracy, we refer to and improve the fitness function construction strategy adopted by Zhang et al. [35] in the evolutionary algorithm, which can take into account classification accuracy and dimensionality reduction.

$$\begin{cases} f(x_i) = \rho \cdot CA + (1 - \rho) \cdot DR \\ \rho = \frac{9 + 0.99\left(\frac{50}{\varphi + 50}\right)}{10} \end{cases}$$

where $CA$ represents classification accuracy and $DR$ represents dimensionality reduction rate. When the feature dimension is small, the value of $\rho$ is close to 1, and the inspection of the fitness function is mainly based on classification accuracy. As the feature dimension increases, the value of $\rho$ decreases until it is close to 0.9, and the dimensionality is reduced. The importance has also increased. Therefore, under the new fitness function, the algorithm considers both the classification accuracy of the model and the reduction of feature dimensions, but the classification accuracy is still the most important.

$$\begin{cases} CA = NCC/NAS \\ DR = 1 - (NSF/NAF) \end{cases}$$

where $NCC$ is the number of samples correctly classified, and $NAS$ is the total number of training samples. $NSF$ represents the dimension of the feature subset after feature selection, and $NAF$ represents the feature dimension.

The measurement of classification accuracy uses K-Nearest Neighbors (KNN). The classification principle of the K-nearest neighbor classifier is to consider that the samples with similar distances in the feature space have the same category. In order to avoid the influence of local outliers, K-nearest neighbor points are selected, and the center point category is selected as the category with the highest proportion among the neighbor points [41]. The establishment of the classification model is "lazy". With the import of testing data, the classification model is constantly being improved. The key parameters in the classification model are K value and distance measurement. The choice of K value is generally determined empirically. In order to reduce the classification time, the K value in this study is selected as 5. The distance metric is used to calculate the distance between sample features, describe the degree of similarity between each sample, and the commonly used Euclidean distance is used in the study. $X_1 = (x_{11}, x_{12}, \ldots, x_{1n})$ and $X_2 = (x_{21}, x_{22}, \ldots, x_{2n})$ are two samples, where $n$ is the number of features, and the distance between two samples is expressed as:

$$dist(X_1, X_2) = \left( \sum_{i=1}^{n} |x_{1i} - x_{2i}|^p \right)^{1/2}$$

The improvement of the feature selection strategy is actually the binarization of the equilibrium concentration value C. In feature selection work, binary coding is often used to eliminate or select features, so here we will use a uniform random number between 0 and 1 to initialize the initial concentration value of each individual. We set the threshold to 0.5, and use the Formula (3) to convert continuous concentration values into discrete values, which can be used for binary coding for feature selection. Since the initialization value is random, the probability of each feature being selected and the probability of being eliminated are equal. After the Equalization Optimizer algorithm uses the optimal individual to update the concentration value of each individual, there will be a concentration value greater than 1 or less than 0. Although they have no effect on the feature selection process, this error will continue to accumulate in subsequent iterations. The equilibrium concentration values greater than 1 or less than 0 are corrected to 1 and 0 at the end of each iteration.

$$SF\_bin = \begin{cases} feature\ selected, & If\ 0.5 \leq C_t^d \\ feature\ net\ selected, & If\ 0.5 > C_t^d \end{cases} \tag{3}$$

In terms of parameter settings, we set the number of iterations to 100 and the number of individuals to 100. Since there are only 8000 training samples, 5-fold cross-validation is used to obtain a relatively stable classification result of the KNN classifier.

### 3.4. Classification and Accuracy Evaluation

SVM is an efficient machine learning classification method. It obtains the support vector on the category boundary and divides the optimal boundary of the classification according to the support vector. At present, the robustness of SVM has been proven in text classification [42], image classification [43], biometric recognition [44] and other fields, and it is also very commonly used in land cover classification [45–47]. SVM does not have high requirements for training samples, and good classification results can be obtained by only using small samples for training. Point cloud category labeling is very difficult, and a large number of training samples are difficult to obtain. SVM classifier can adapt to the training set of small samples, so it is researched to apply SVM to the land cover classification of the Titan multispectral LiDAR point cloud. After testing, the polynomial kernel function was chosen, coef0 was set to 8, and gamma was set to $-0.0625$.

The classification result evaluation is used to measure the effectiveness of feature construction and the applicability of feature selection. The confusion matrix can provide the classification effect of each category in the classification process. This study uses the confusion matrix generated by the classification result and uses the confusion matrix to calculate producer's accuracy (PA), user's accuracy (UA), overall accuracy (OA), class average accuracy (AA) and kappa coefficient. Here, $i$ represents the category of ground objects, and $a_i$ is the correct classification number of category $i$. $x_i$ and $y_i$ is the number of category $i$ ground objects samples in the ground truth label and predicted label, respectively. $N$ is the total number of samples and $C$ is the number of categories.

$$PA_i = \frac{a_i}{x_i}$$

$$UA_i = \frac{a_i}{y_i}$$

$$OA = \frac{\sum_{i=1}^{8} a_i}{N}$$

$$AA = \frac{\sum_{i=1}^{8} PA_i}{C}$$

$$Kappa = \frac{N \sum_{i=1}^{C} a_i - \sum(x_i y_i)}{N^2 - \sum(x_i y_i)}$$

## 4. Result

In order to verify the effectiveness of the proposed land cover classification method and explore the role of neighborhood features and feature selection in this method, a series of comparative experiments are set up (Table 3).

**Table 3.** Experimental cases and classification features.

| Case | Classification Feature |
|---|---|
| Case 1 | 3 wavelength spectral information + height information |
| Case 2-1 | Single-scale neighborhood feature (K = 20) |
| Case 3-1 | Single-scale neighborhood feature (K = 50) |
| Case 4-1 | Single-scale neighborhood feature (K = 100) |
| Case 5-1 | Single-scale neighborhood feature (K = 150) |
| Case 6-1 | Multi-scale neighborhood feature (K = 20, 50, 100, 150) |
| Case 2-2 | Single-scale neighborhood feature (K = 20) + feature selection |
| Case 3-2 | Single-scale neighborhood feature (K = 50) + feature selection |
| Case 4-2 | Single-scale neighborhood feature (K = 100) + feature selection |
| Case 5-2 | Single-scale neighborhood feature (K = 150) + feature selection |
| Case 6-2 | Multi-scale neighborhood feature (K = 20, 50, 100, 150) + feature selection |

### 4.1. Classification Results of Neighborhood Features

#### 4.1.1. Classification Results of Single-Scale Neighborhood Features

We extracted neighborhood features at small, medium, and large scales (K = 20, 50, 100, 150), and compared the classification results with the original information (Case 1). The original information has four dimensions, including three-channel spectral information and height information. Neighborhood features have 43 dimensions, including neighborhood spectral features and neighborhood spatial features. Under the condition of only using small samples to train the classifier, using single-scale neighborhood features can obtain more than 80% of OA and more than 84% of AA (Table 4). Compared with the original information classification (Case 1), OA and AA increased by 4.3–14% and 0.06–7.8%, respectively, and the kappa coefficient increased by 0.05–0.17. This result shows the advantages of neighborhood features. It can use the information of neighborhood points to reduce classification errors caused by outliers in the data acquisition process. From a visual

point of view (Figure 4), neighborhood features can significantly reduce salt and pepper noise. And as the neighborhood scale increases, the effect of reducing salt and pepper noise becomes more significant. In Case 1 of Figure 4, there are a large number of Trees classified as Cars. It can be found in Case 2-1 to Case 5-1 that this type of error is almost eliminated.

**Table 4.** Comparison of classification accuracy between original information and single-scale neighborhood features.

|  | Case 1 | Case 2-1 | Case 3-1 | Case 4-1 | Case 5-1 |
|---|---|---|---|---|---|
| Feature dimension | 4 | 43 | 43 | 43 | 43 |
| OA | 76.33% | 80.65% | 85.89% | 88.68% | 90.36% |
| AA | 84.74% | 84.80% | 89.03% | 91.40% | 92.54% |
| Kappa | 0.705 | 0.753 | 0.8169 | 0.8519 | 0.8732 |

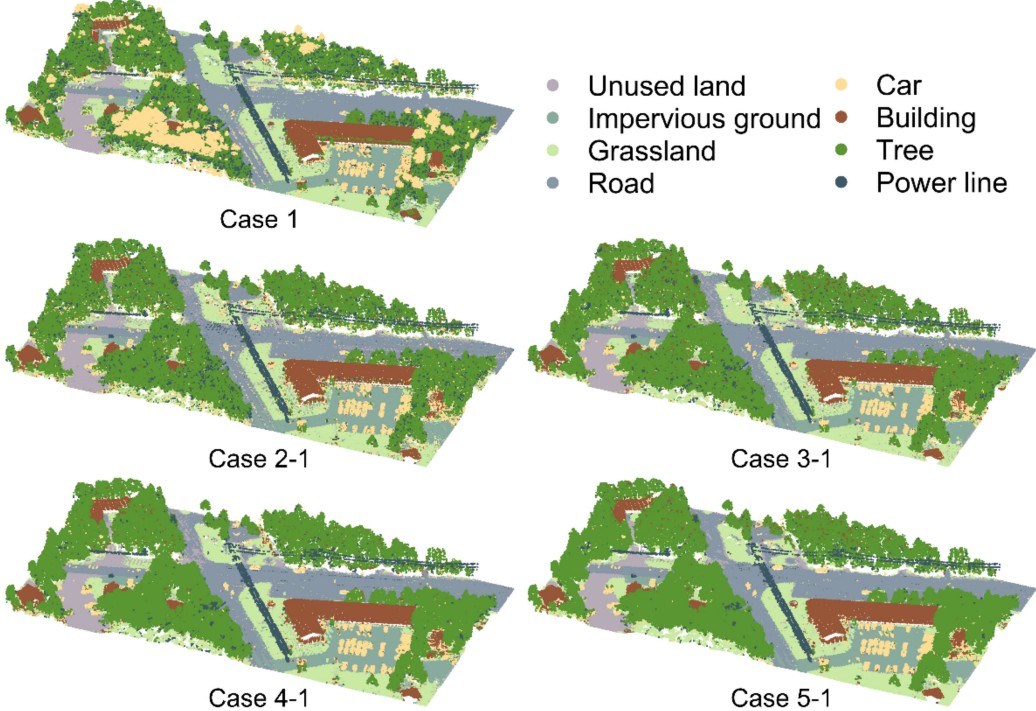

**Figure 4.** Comparison of original information and single-scale neighborhood feature classification results.

### 4.1.2. Classification Results of Multi-Scale Neighborhood Features

Compared with the original information, the single-scale neighborhood feature has an outstanding performance. However, we still recognize the limitations brought by the single-scale neighborhood feature. The sizes of the objects in the scene are different, and the boundaries between different objects are difficult to deal with. Small-scale neighborhood features (Case 2-1) can better deal with the boundary between different objects. The large-scale neighborhood feature (Case 5-1) is more effective for the misclassification points inside the objects. Some examples are given in Figure 5, and the subfigures (a)–(c) all confirm this conjecture. In order to obtain neighborhood features with better classification performance, we concatenate four single-scale neighborhood features. Finally, we obtained 160-dimensional multi-scale neighborhood spatial–spectral features.

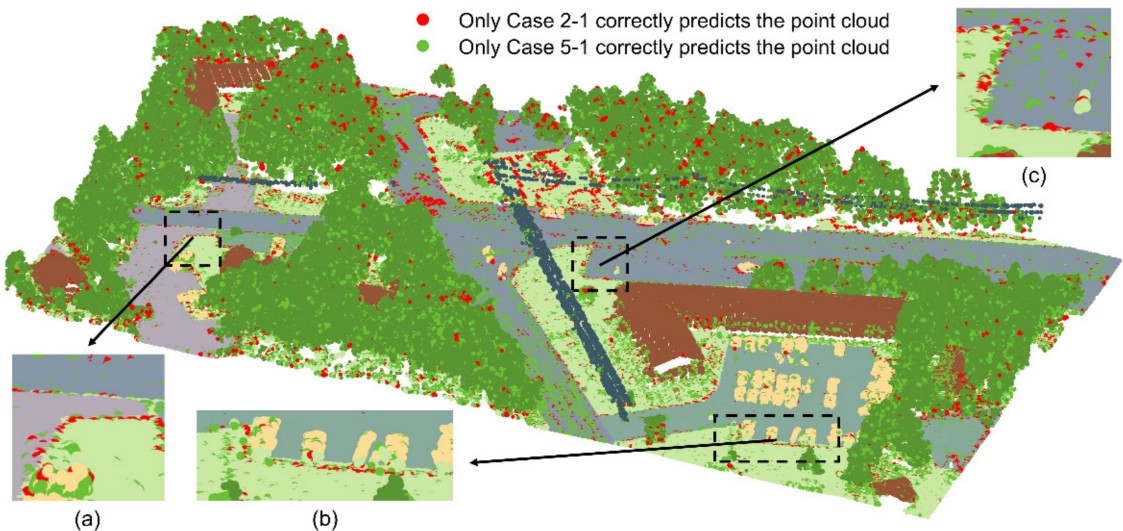

**Figure 5.** Comparison of classification results between Case 2-1 and Case 5-1. The red marked point cloud is the point correctly classified in Case 2-1, but not correctly classified in Case 5-1. The point cloud marked in green is the point correctly classified in Case 5-1, but not correctly classified in Case 2-1. Refer to Figure 4 for the legend of objects in this figure. The subfigure (**a**–**c**) shows the details.

Surprisingly, the multi-scale neighborhood features did not achieve better classification accuracy, but the accuracy was greatly reduced. After the feature concatenation, the performance of the neighborhood feature has "degraded". Its OA and AA are 72.39% and 78.84%, respectively, which are even 4.0% and 5.9% lower than the original information classification results. The reason for this phenomenon is that the feature dimension is too high, there is more redundant information, and 'Hughes' phenomenon occurs. In order to solve this problem, we also adopted further feature selection for dimension reduction.

### 4.2. Feature Selection Based on Equalization Optimizer Algorithm

#### 4.2.1. Feature Subset

We applied feature selection based on Equalization Optimizer optimization algorithm to four single-scale neighborhood features and one multi-scale neighborhood feature. There is no "Hughes" phenomenon in single-scale neighborhood features. However, to verify the effectiveness of multi-scale neighborhood features, we use single-scale neighborhood features as a control experiment. Table 5 indicates that the feature dimensions of all neighborhood features are compressed to a small range. Due to the characteristics of the designed fitness function, the higher the feature dimension, the greater the proportion of optimization for the purpose of dimensionality reduction. So, the 160-dimensional multi-scale feature has the highest dimensional compression rate. The dimension reduction rate of single-scale features is the same. When the Equalization Optimizer algorithm is used for feature selection, the content of the obtained feature subset is not stable, but the dimensionality reduction rate and the classification accuracy of the application of the feature subset for classification remain basically stable.

By analyzing the content of the feature subset, we found that in all the feature subsets: the spectral feature had been selected 29 times and the spatial feature had been selected 30 times. Among them, the feature types of the spectral features are relatively concentrated, with $mean^K{}_j$, $CV^K{}_j$, $ratio^K{}_j$, $NDFI^K{}_{G-NIR}$ selected many times, and the spatial features are relatively scattered, with $Z$, $Std^K z$, $\Delta z^K$, $R^K$, $C^K{}_\lambda$, $O^K{}_\lambda$, $S^K{}_\lambda$ selected many times. In the spatial feature, $Z$ and $\Delta z^K$ were selected every time, and it can be considered that the two have a better ability to distinguish ground objects. Due to the combination effect between the various features, it is not possible to conclude here that a certain feature is important to the Titan multispectral LiDAR land cover classification work. However,

the above combination of features has instructive significance for the construction of classification features.

**Table 5.** Dimension reduction rate and feature subset of neighborhood feature after feature selection.

| Neighborhood Size | Feature Dimension | | Dimension Reduction Rate | Feature Subset |
|---|---|---|---|---|
| | Before Feature Selection | After Feature Selection | | |
| Case 2 | 43 | 11 | 74.41% | $mean^{20}{}_1, mean^{20}{}_2, mean^{20}{}_3, CV^{20}{}_2, CV^{20}{}_3, ratio^{20}{}_3,$ $e^{20}{}_3, R^{20}, \Delta z^{20}, Std^{20}z, Z$ |
| Case 3 | 43 | 11 | 74.41% | $mean^{50}{}_2, std^{50}{}_{dev_3}, CV^{50}{}_2, ratio^{50}{}_1,$ $NDFI^{50}G-NIR, O^{50}{}_\lambda, R^{50}, V^{50}{}_\lambda, \Delta z^{50}, Std^{50}z, Z$ |
| Case 4 | 43 | 11 | 74.41% | $mean^{100}{}_2, mean^{100}{}_3, CV^{100}{}_2, CV^{100}{}_3, ratio^{100}{}_2,$ $NDFI^{100}{}_{G-MIR}, O^{100}{}_\lambda, C^{100}{}_\lambda, R^{100}, \Delta z^{100}, Z$ |
| Case 5 | 43 | 11 | 74.41% | $mean^{150}{}_1, mean^{150}{}_2, CV^{150}{}_1, ratio^{150}{}_2, NDFI^{150}{}_{G-NIR},$ $S^{150}\lambda, O^{150}{}_\lambda, R^{150}, \Delta z^{150}, Std^{150}z, Z$ |
| Case 6 | 160 | 15 | 90.63% | $mean^{50}{}_1, S^{50}{}_\lambda, \Delta z^{50}, Std^{50}z, CV^{100}{}_2, C^{100}{}_\lambda, Std^{100}z,$ $std^{150}{}_{dev_1}, kurtosis^{150}{}_3, ratio^{150}{}_3, NDFI^{150}{}_{G-NIR},$ $S^{150}{}_\lambda, E^{150}{}_\lambda, C^{150}{}_\lambda, Z$ |

### 4.2.2. Classification Results of Multi-Scale Neighborhood Feature Subset

We performed classification tests on the feature subsets obtained by feature selection based on the Equalization Optimizer algorithm and obtained OA and AA (Table 6) for each case. The results show that after the feature selection of the five groups of neighborhood features through the Equalization Optimizer algorithm, except for the neighborhood features at the K = 150 scale, the remaining four groups obtained better results. The OA and AA of single-scale neighborhood feature classification results increased by 0.08–4.68% and 0.15–1.88% compared with those before feature selection. The most obvious improvement is the multi-scale neighborhood feature, which eliminates the "Hughes" phenomenon (Figure 6) caused by high dimensions. The OA and AA of the multi-scale neighborhood feature classification result increased by 19.6% and 14.57%, respectively, and performed best among the five different types of neighborhood features tested. The confusion matrix of the classification result is shown in Figure 7.

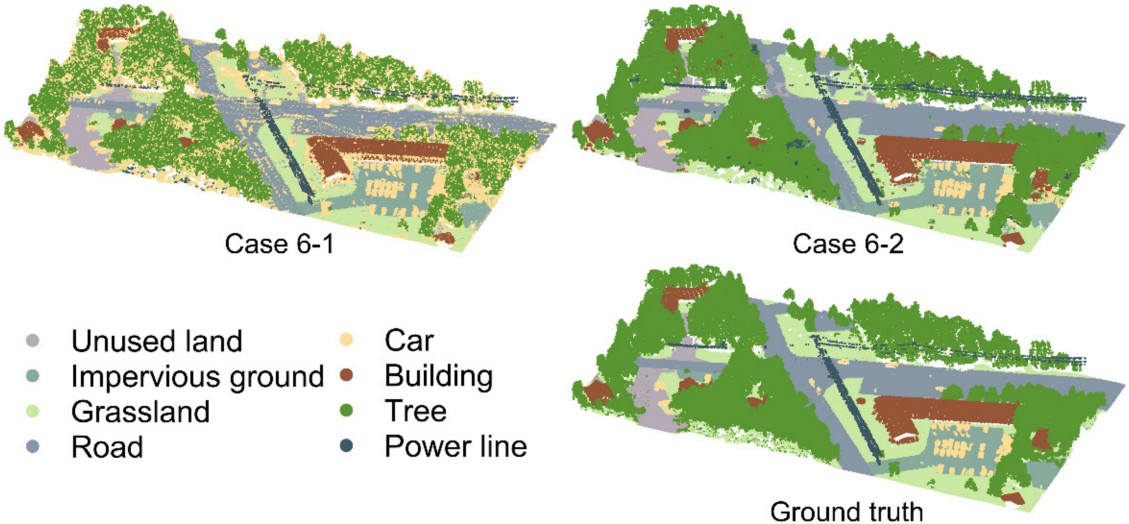

**Figure 6.** Comparison of multi-scale neighborhood feature classification results before and after feature selection.

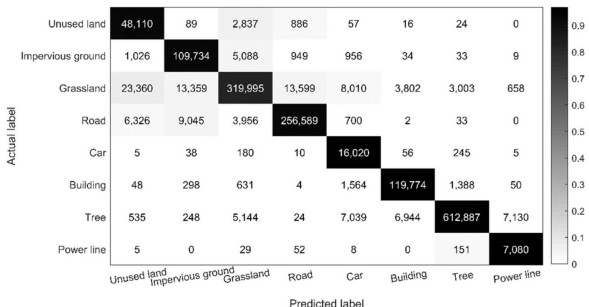

**Figure 7.** Confusion matrix of multi-scale neighborhood feature subset (Case 6-2) classification result (the background color is rendered with PA).

**Table 6.** Classification accuracy of feature subset after feature selection.

|  | Case 2-2 | Case 3-2 | Case 4-2 | Case 5-2 | Case 6-2 |
|---|---|---|---|---|---|
| Feature dimension | 11 | 11 | 11 | 11 | 15 |
| OA | 85.33% | 89.55% | 89.90% | 90.44% | 91.99% |
| AA | 86.68% | 90.90% | 91.55% | 91.98% | 93.41% |
| Kappa | 0.8095 | 0.8628 | 0.8673 | 0.8742 | 0.8942 |

## 5. Discussion

To explore the reliable land cover classification method of Titan multispectral LiDAR point cloud data, this study constructed 43 spatial–spectral features. The spatial features are mainly constructed with reference to the features in single-wavelength LiDAR target classification research [28]. The spectral features use the statistical features of the three-channel spectral intensity and NDFI. The NDFI is designed with reference to NDVI for Titan Multispectral LiDAR point cloud data [16,22,23], and has an excellent performance in the classification of vegetation and buildings. In this study, the spatial–spectral features were constructed based on the spatial location of the point cloud, combined with the neighborhood points of each central point cloud. To highlight the advantages of the neighborhood feature, we compared the classification result of the neighborhood feature with the classification result of the original information (three-channel spectral pseudo reflectance + height value). Except that the classification effect of multi-scale neighborhood features, which caused the "Hughes" phenomenon due to the high feature dimension, was worse than the original information, the classification results of single-scale neighborhood features were significantly better than the original information. The characteristic difference shown by the original information is isolated; that is, the spatial information and the spectral information do not have a good synergy. Using the local spatial position of the point cloud to construct neighborhood points, and locally extracting the spectral and spatial features of the multispectral point cloud can give full play to the potential of spatial–spectral combination. In the process of acquiring multispectral point cloud data, there are some abnormal values, which will aggravate the "salt and pepper" phenomenon in the classification results. Using the spatial–spectral information of neighboring points can greatly reduce this phenomenon. Additionally, in the studied neighborhood scale, as the number of neighborhood points increases, the elimination of this phenomenon is more obvious. This conclusion has been shown in Figure 8. Among the neighborhood features constructed in this research, as the K value continues to increase, the classification effect of single-scale neighborhood features is better. Whether it is OA, AA, or PA and UA for each feature category, they all comply with this rule. However, since the larger the K value, the more the number of neighborhood points, the higher the calculation cost, this study did not continue to increase the number of neighborhood points. However, it can still be inferred that as the size of the neighborhood increases, the types of features contained in the neighborhood points tend to become more complicated, and the performance of

the extracted spatial–spectral features will decrease. The classification accuracy does not continue to increase as the size of the neighborhood increases.

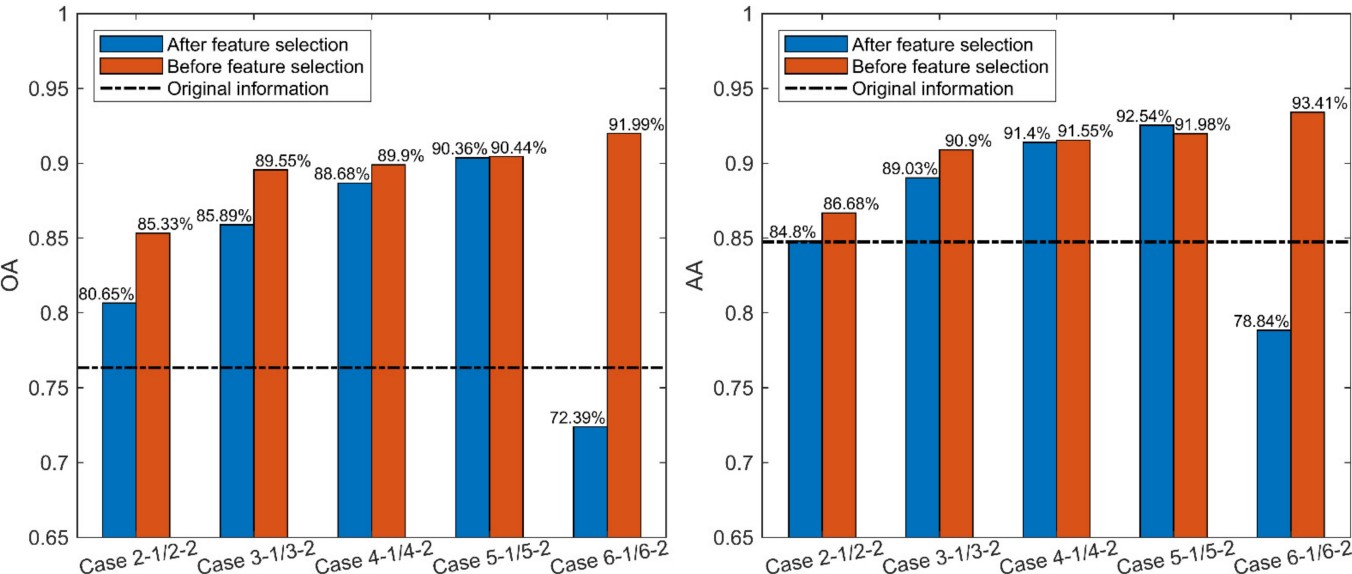

**Figure 8.** Comparison of classification accuracy of original information, single-scale neighborhood features and multi-scale neighborhood features. The original information includes three-channel spectral pseudo reflectance and height information; multi-scale neighborhood features include four single-scale neighborhood features (k = 20, 50, 100, 150).

In the K value scheme in this study, a larger neighborhood scale can achieve higher classification accuracy. This is because the distribution of the objects is continuous, and the large-scale neighborhood has a stronger smoothing effect, and a better classification effect can be obtained in the interior area of the objects. However, as the neighborhood scale becomes larger, more local details are ignored, resulting in misclassification of the boundary area. Large-scale neighborhoods cannot obtain good classification performance at the boundaries of objects, while small-scale neighborhoods are easier to capture detailed information. Therefore, it is necessary to consider the coordination of neighboring features at different scales. There are methods for extracting features by adaptive neighborhood size [27,31,32], but the extracted features are all single-scale. It is only the optimal scale of spatial features, and there is no relevant research on the Titan multispectral LiDAR point cloud. Therefore, this study constructed a multi-scale neighborhood feature to explore the applicability of multi-scale neighborhood features in land cover classification. However, the concatenation of multiple single-scale neighborhood features leads to the high-dimensionality of multi-scale neighborhood features. Under the condition of using only a small sample training set, high-dimensional features have a "Hughes" phenomenon in the SVM classifier, which leads to a worse classification result of multi-scale neighborhood features than any single-scale neighborhood feature. It can be inferred that this is because there is a lot of redundant information or interference information in the multi-scale neighborhood features. To exert its true classification performance, it is necessary to screen the features, so the research considers the feature selection.

The research uses the Equalization Optimizer algorithm to perform feature selection on five kinds of neighborhood features and conducts a classification test on the obtained five feature subsets. The test results intuitively reflect the effectiveness of feature selection (Figure 8). It is worth pointing out that the use of feature selection based on the Equalization Optimizer algorithm solves the "Hughes" phenomenon that occurs when the feature dimension is too high in the multi-scale neighborhood. The classification performance of multi-scale neighborhood feature subsets has been significantly improved, which can take advantage of the advantages of multi-scale neighborhood features. Whether com-

pared with single-scale neighborhood features or compared with multi-scale neighborhood features without feature selection, multi-scale neighborhood feature subsets have strong competitiveness. From the perspective of PA and UA of a single feature category (Table 7), the accuracy performance of the multi-scale neighborhood feature subset (Case 6-2) is satisfactory and very stable. The parts in bold in Table 7 highlight the best accuracy for each object type. Under the premise of ensuring that the classification accuracy is not affected, the feature selection work can remove redundant information, greatly reduce the dimension of the feature, and obtain a better classification effect with the lowest computational cost. Thanks to the design of the fitness function, the fitness value reasonably balances the relationship between classification accuracy and feature dimensions. When dealing with high-dimensional features, it can give full play to the role of dimensionality reduction and has strong feature compression capabilities. When dealing with low-dimensional features, it better improves the accuracy of classification.

**Table 7.** Comparison of producer accuracy (PA) and user accuracy (UA) of neighborhood feature subset.

|  | Case 2-2 | | Case 3-2 | | Case 4-2 | | Case 5-2 | | Case 6-2 | |
|---|---|---|---|---|---|---|---|---|---|---|
|  | PA | UA | PA | UA | PA | UA | PA | UA | PA | UA |
| Unused land | 87.19% | 50.17% | 89.75% | 55.48% | 90.97% | 55.02% | 92.09% | 53.08% | **92.49%** | **60.58%** |
| Impervious ground | 87.15% | 73.82% | 90.41% | 80.42% | 91.55% | 78.35% | 90.70% | 81.65% | **93.13%** | **82.62%** |
| Grassland | 75.31% | 92.52% | 80.50% | 92.98% | 79.44% | 93.91% | 80.58% | 94.13% | **82.95%** | **94.71%** |
| Road | 86.65% | 91.83% | 90.76% | 94.08% | 90.21% | 93.13% | 90.30% | 94.02% | **92.75%** | **94.30%** |
| Car | 85.89% | 21.32% | 92.37% | 34.90% | 93.70% | 37.97% | 96.04% | 37.41% | **96.74%** | **46.63%** |
| Building | 90.76% | 89.39% | 94.42% | 92.20% | 95.96% | 91.00% | 95.76% | **94.36%** | **96.78%** | 91.69% |
| Tree | 89.17% | 98.43% | 93.22% | 98.93% | 94.34% | 99.34% | 95.03% | 99.14% | **95.77%** | **99.21%** |
| Power line | 91.36% | 15.50% | 95.77% | 24.39% | 96.22% | 30.91% | 95.32% | 32.90% | **96.66%** | **47.41%** |

The parts in bold in Table highlight the best accuracy for each object type.

Compared with previous work, the focus of our research is to fully excavate the spatial–spectral information of multispectral LiDAR point cloud data and obtain a point-based land cover classification method with small sample training. Large-scale land cover classification based on multispectral LiDAR data is time-consuming and labor-intensive. In order to obtain a more convenient processing flow and lower calculation cost, most of the research has carried out with an image-based method [18,21,22,24]. They achieved an overall accuracy of around 90% or even higher. However, existing studies have shown that point-based classification is better than image-based classification [19]. In addition, under the point-based classification method, there are more categories (such as power lines, cars, under-forest areas, etc.). More refined classification will increase the overall difficulty of classification, so we believe that the classification accuracy obtained by comparing the two classification methods is of no practical significance. In the point-based classification study, Xiao et al. [48] carried out 3D land cover classification based on an Object Based Image Analysis (OBIA) method, and obtained similar accuracy (OA, 91.63%; kappa 0.895) to this study (OA, 91.99%; kappa, 0.8942). Compared to the feature threshold classification method, our proposed method has fewer parameter settings (only K value and classifier parameters), and has better robustness in different scenarios. Ekhtari et al. [19] processed single-return points and multi-return points separately, and obtained higher accuracy (OA, 94.7%; kappa, 0.94), especially in the Building and Grassland categories, to obtain a lower classification error. The different number of training samples may be the cause of the difference in accuracy. In order to explore the feasibility of small sample training, we only used 0.5% of the total sample as the training set, while they used 30% of the total sample. In addition, the effectiveness of our method should be explored in a wider range of scenarios in the future to meet the actual needs of land cover classification.

## 6. Conclusions

Titan multispectral LiDAR provides a wealth of spectral and spatial information. By fully mining various spatial–spectral features, it can greatly improve the performance of multispectral LiDAR point clouds in land cover classification. We propose a land

cover classification method suitable for Titan multispectral LiDAR point clouds. This study uses K-nearest neighborhoods to extract the spectral and spatial features of single-scale multispectral LiDAR point clouds in small, medium, and large-scale neighborhoods (K = 20, 50, 100, 150). Multi-scale neighborhood features are obtained through feature serialization, which makes up for the lack of spatial–spectral collaboration information in the original information. In addition, we used the Equalization Optimizer algorithm to perform feature selection on high-dimensional multi-scale neighborhood features and obtain low-dimensional multi-scale neighborhood features with stronger classification performance. Finally, the SVM classifier is used for classification. In the case of using only small training samples, 91.99% OA, 93.41% AA and 0.89 kappa coefficient were obtained in the study area. Compared with the original information, OA, AA and kappa coefficient increased by 15.66%, 8.7% and 0.19, respectively.

By comparing the classification results of the original information, single-scale neighborhood features, multi-scale neighborhood features, single-scale neighborhood feature subsets obtained by feature selection, and multi-scale neighborhood feature subsets, we get the following conclusions:

- Compared with the original information, the 43 spatial–spectral features constructed by this research have significant advantages in land cover classification. Feature extraction based on neighborhood points can play a synergistic effect of spatial information and spectral information and play a key role in land cover classification.
- The scale of the neighborhood has an impact on the classification performance of features. Small-scale neighborhoods have better classification performance in the boundary area of the features, and large-scale neighborhoods improve the classification accuracy of the interior areas of the features more significantly.
- The multi-scale neighborhood features obtained by concatenating the features of single-scale neighborhoods will cause the "Hughes" phenomenon due to the high dimensionality. Only concatenating single-scale neighborhood features cannot give full play to the advantages of multi-scale.
- Feature selection based on the Equalization Optimizer algorithm can obtain feature subsets with lower dimensions and better performance. Compressing the 160-dimensional multi-scale neighborhood feature to 15-dimensional, the overall accuracy is improved from 72.39% to 91.99%; thus solving the "Hughes" phenomenon caused by the high dimensionality.

**Author Contributions:** S.S. (Shuo Shi) and B.C. (Biwu Chen) provide research directions and guidance; F.Q. provides help with experimental data; S.B. designs and conducts experiments to complete the writing of the paper; W.G., S.S. (Shuo Shi), S.S. (Shalei Song), B.C. (Bowen Chen) and X.T. provide guidance and editing for the writing of the paper. All authors have read and agreed to the published version of the manuscript.

**Funding:** This work was supported by the National Key R&D Program of China (2018YFB0504500); the National Natural Science Foundation of China (41971307); China Postdoctoral Science Foundation (2017T100582); and LIESMARS Special Funding.

**Institutional Review Board Statement:** Not applicable.

**Informed Consent Statement:** Not applicable.

**Data Availability Statement:** No new data were created or analyzed in this study. Data sharing is not applicable to this article.

**Acknowledgments:** The authors would like to thank the ISPRS Commission III and WG III/5 for providing the raw data of the Harbor of Tobermory data set.

**Conflicts of Interest:** The authors declare no conflict of interest.

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
