# Peer review of "Land Cover Classification with Multispectral LiDAR Based on Multi-Scale Spatial and Spectral Feature Selection"

_remotesensing, doi:10.3390/rs13204118_

Round 1

Reviewer 1 Report

Review of the manuscript ID remotesensing- 1405699 entitled: Land Cover Classification with Multispectral LiDAR based on Multi-scale Spatial and Spectral Feature Selection
In the presented study, the Authors address addresses the problem of improving the efficiency of multi-spectral LiDAR data use. The authors propose a land cover classification method based on multi-scale spatial and spectral feature selection. Tests in 8 manually separated categories were performed on a publicly available dataset acquired with an Optech Titan Multispectral airborne laser scanner. The method includes a 4-step classification process: 1) K-nearest point selection, 2) spatial-spectral feature extraction, 3) feature selection, and 4) classification using a Support Vector Machine (SVM) classifier. The use of machine learning in this process significantly improved the classification efficiency from 9 to 16%, thus confirming the high efficiency of the Equalizer Optimization algorithm for feature selection. The results obtained confirm the utility of the method proposed by the authors for land cover classification based on point data of multispectral LiDAR system and highlight the importance of selecting an appropriate perform feature selection. The conducted study also has a practical dimension and may contribute to better land use planning and management.
In my opinion, the study itself as well as the obtained results and their interpretation do not raise any doubts. The layout of the paper is clear and the results was plausible in the validation process. I note the minor shortcomings of this work. First, I miss the flowchart showing all the steps of the method used. The authors should include this flowchart in the method chapter. Second, the discussion is perhaps too little oriented towards referring to the results of other studies. It is extensive section, but poorly referenced to the ‘state of the art’. There are only 3 references in this section, in addition all of them concern the same thread - methods for extracting features by adaptive neighbourhood size. This chapter should be reconsidered by the authors, and their findings discussed with the achievements of other authors. Other than these minor changes and additions, the manuscript is likely to find great interest among Remote Sensing Journal readers.

Author Response

Thank you very much for your comments and replies. We have arranged the reply and the updated contents of the manuscript in the attachment. Please see the attachment.

Looking forward to your reply again.

Reviewer 2 Report

This paper is very well presented and motivated. The authors explained how the sample was taken, processed and evaluated with different algorithms. The algorithm proposed has very good preformance but it is not constrasted with other methods. The experimental evaluation just picks different scenarios and shows how the algorithm provides a good response.

This is just an style comment, equations are numbered but never referenced by their numbers so the numeration can be removed, just 24, 25, 26 and 29 I think are referenced.

Table 4 has something wrong with the formatting and it would probably be solved during the edition process if the paper is accepted.

Caption for Figure 3 is in a different page.

Author Response

(The authors gave the same response as above.)

Reviewer 3 Report

Remote Sensing.

Review Report

Manuscript remotesensing-1405699. “Land Cover Classification with Multispectral LiDAR based on Multi-scale Spatial and Spectral Feature Selection.”

The manuscript is about a proposal of a method of classification of the land cover based on the selection of spatial and spectral characteristics of multiple scales, applying the public data set of the Tobermory port collected by the Optech Titan Multispectral aerial laser scanner.

It is an interesting work that is innovative when combining spectral and spatial information in the Land Cover Classification process. The work is well supported, well structured, and well presented.

HOWEVER, I FIND A FEW MINIMUM AREAS OF OPPORTUNITY THAT CAN BE ADJUSTED TO IMPROVE THE WORK IN ITS CURRENT FORMAT.

GLOBAL COMMENTS

I find that the English language and style are acceptable, and only a minor spell check is required

No annotation is made that distinguishes Land Use and Land Cover (LULC); Instead, “Land Cover” is used without any distinction and referring to both when they are different things, as can be confirmed in many recent works. Even in the categories selection of your work, some elements do not correspond to “Land Cover.”

Vivekananda, G. N., Swathi, R., & Sujith, A. V. L. N. (2021). Multi-temporal image analysis for LULC classification and change detection. European journal of remote sensing, 54(sup2), 189-199.

da Silva, V. S., Salami, G., da Silva, M. I. O., Silva, E. A., Monteiro Junior, J. J., & Alba, E. (2020). Methodological evaluation of vegetation indexes in land use and land cover (LULC) classification. Geology, Ecology, and Landscapes, 4(2), 159-169.

Pramit, V., Aditya, R., Srivastava, P. K., & Raghubanshi, A. S. (2020). Appraisal of kappa-based metrics and disagreement indices of accuracy assessment for parametric and nonparametric techniques used in LULC classification and change detection. Modeling Earth Systems and Environment, 6(2), 1045-1059.

Of course, it is possible to refer only to “Land Cover” for convenience in your work, but I think it is necessary to clarify it before doing so, except for your best opinion.

Below, I provide a few detailed comments for the authors’ consideration which I hope they will see positively in further improving their work.

DETAILED COMMENTS

  1. In line 146, you mention “... image of Google map ...” apparently you refer to the Google Maps platform, and then you must use the correct name.

  1. in topic 2.3 Training and testing samples. (Lines 191-199)

You mention that 1000 samples were randomly selected for each of the 8 categories in your study as training points for the classification algorithm.

It is also mentioned that 1,627,877 point clouds were obtained, and all of them were used as testing samples to evaluate the precision of the results.

Here I have a doubt. Does this last mean that the 1,627,877 points (including the 8000 samples used in the training stage) are the Ground Truth sample to evaluate the accuracy of the maps obtained?

If this is so, I think that at least the 8000 points used in training should be omitted from the evaluation, and they cannot be both “judge and part.”

  1. I think not all variables in equations 30-33 are described.

  1. The confusion matrix is mentioned in the final part of the methodology (line 397), but it is not included in the results. I think including the confusion matrices (or some of them) would help a lot to see the behavior in detail of the classification results, both Globally and for each category.

  1. Optionally, as many classification works do, it may be convenient to evaluate the inclusion of the Kappa Concordance Coefficient as one more accurate metric.

Author Response

(The authors gave the same response as above.)
